# Developing a competency assessment framework for medical laboratory technologists in primary healthcare settings in India

**Sanjeev Kumar**[1]*, **Gaurav Chhabra**[2], **Kaptan Singh Sehrawat**[3,4], **Malkit Singh**[5]

1 Health Systems Transformation Platform, New Delhi, India, 2 Department of Pathology and Laboratory Medicine, All India Institute of Medical Sciences, Bhubaneswar, Odisha, India, 3 Indian Confederation of Medical Laboratory Science, New Delhi, India, 4 Department of Biochemistry Kalawati Saran Children's Hospital, New Delhi, India, 5 Department of Medical Microbiology, Post Graduate Institute of Medical Education and Research, Chandigarh, Punjab, India

* snjvkumar386@gmail.com, Sanjeev.kumar@hstp.org.in

**Data Availability Statement:** All relevant data are within the paper and its Supporting Information files.

## Abstract

Medical Laboratory Technologists play a significant role in delivering quality laboratory Services. The competency assessment of MLTs is a critical driver for enhancing primary healthcare performance. While several countries have developed competency frameworks for MLTs in primary care, such frameworks are lacking in the Indian context. This study aimed to create a competency assessment framework to assess the competencies of MLTs (Medical Laboratory Technologists) working in Indian public primary healthcare facilities. The research followed a five-step process, starting with a review of existing literature on MLTs' competencies in primary healthcare. Expert consultations were then conducted to establish a consensus on these competencies. Following this, assessment tools were developed based on the literature review and expert input. Another round of expert consultations was held to ensure agreement on the assessment tools. Finally, the developed tools were tested in a public primary healthcare facility. The literature review identified 86 competencies across 11 domains: safe work practices, data/ sample collection, specimen preparation equipment instruments and regiments, assessment and analysis, recording and reporting, infection control, quality management, critical thinking, communication and interaction, and professional practice. Expert consultations resulted in the consensus on ninety-five competencies in ten domains of MLTs in primary healthcare settings. Competencies for each domain were discussed and agreed upon. A competency assessment tool was finalized after unanimous agreement among experts. The competency assessment tool was later finalized after pre-testing on MLTs in a clinical laboratory part of a public primary health care facility. This study successfully developed a competency assessment framework for in-service MLTs in Indian public primary healthcare settings. The framework encompasses ninety-five competencies covering ten domains of MLT responsibilities. It provides a comprehensive tool for assessing MLT's competencies and identifying competency gaps. The framework can be used to capacitate MLTs, improve their performance in primary healthcare settings, and enhance the delivery of healthcare services in India. It bridges a critical

**Funding:** This study was supported by the Health Systems Transformation Platform, New Delhi. The funders had no role in study design, data collection and analysis, decision to publish, or preparation of the manuscript.

**Competing interests:** The authors have declared that no competing interests exist

**Abbreviations:** LMIC, Low- and Middle-Income Countries; MLT, Medical Laboratory Technologists; PHC, Primary health care; WHO, World Health Organization; CAT, Competency Assessment Tool; CBT, Competency-based Training; HSTP, Health Systems Transformation Platform; MSDS, Master Safety Data Sheet; TAT, Turn Around Time; BMW, Bio-Medical Waste; 4 M, Money, Man, Material, and Minutes; SOP, Standard of Procedures.

gap in the existing literature and can aid as a valuable resource for policymakers, educators, and healthcare professionals involved in practicing medical laboratory Services in primary healthcare settings.

## Introduction

The laboratories at the primary health centre play a critical role in improving patient survival. It is because laboratory services support clinicians in providing quality medical care by performing tests and developing evidence-based laboratory diagnosis and screening for diseases [1, 2]. Clinical laboratory tests are crucial in diagnosing the disease, but many tests are of prognostic value, which helps the physician understand whether the patient is responding to the treatment [3]. Relying solely on clinical diagnosis without confirming it through laboratory testing can result in misdiagnosis in many patients. It can also lead to increased mortality and morbidity [4]. Considering the importance of laboratories in primary health Care, at the 32nd World Health Assembly 1979, it was emphasized to give due consideration to laboratory services with appropriate technology. In 1996, the World Health Organization hosted a consultation in South Africa in which the experts reviewed and finalized the minimum list of tests that should be available for primary healthcare in developing countries based on four basic principles [2]. These are.

A. Tests that are clinically useful in influencing diagnosis and management

B. One test rather than two or more tests where one selected test can provide adequate information.

C. Tests that can be performed easily, quickly, and with cheaper reagents, but without compromising reliability, rather than those that need more sophisticated equipment.

D. Screening sets of tests (Profile testing) or only individually specified tests

Precisely, tests or sets of tests for any clinical categories should be considered in terms of clinical utility, technical reliability, and implications for laboratory organization and management.

International Health Regulation (IHR) 2005 lists laboratory services as a core function of health systems [5]. The health system must have integrated lab Services at primary health centers, like- Sustainable Development Goals(SDGs) advocate having integrated laboratory services in primary healthcare to provide universal health care services [6].

In 2008, the World Health Organization (WHO) regional office for Africa organized a consensus meeting in Maputo, Mozambique, focused on harmonizing and standardizing clinical laboratory testing. This meeting led to the establishment of the Maputo Declaration on laboratory systems, which aimed to tackle the challenges in laboratory services that had hindered their expansion [7]. Despite these global efforts, in most countries, laboratory services in primary care are poor [8]. Availability of effective laboratory Services at primary health centers increase the service utilization [9]. Lab workforce competency is one of the key determinants to ensure effective and efficient laboratory services in any settings. Laboratory workforce competencies improve the workforce by providing a guiding framework for producing education and training programs, identifying worker roles and job responsibilities, and assessing individual performance and organizational capacity [10].

Competency is the ability of personnel to apply skill, knowledge, and experience to perform their laboratory duties correctly [11]. Competency assessment is used to certify that laboratory personnel are satisfying their obligations. In the United States, for example, it is required by federal regulations. The following six procedures are the minimal regulatory requirements for the assessment of competency for all personnel performing laboratory testing [12]:

1. Observing the regular performance of tests on patients, including preparation if necessary.

2. Overseeing the handling, processing, and testing of specimens and monitoring the recording and communication of test results.

3. Examining intermediate test results, quality control records, proficiency testing outcomes, and preventative maintenance records.

4. Directly observing the performance of instrument maintenance and functional checks.

5. Evaluating test performance by analyzing previously tested specimens, conducting blind internal tests, or using External Quality Assessment Schemes (EQAS).

6. Assessing problem-solving abilities.

Each test approved by the laboratory director must undergo all six of these procedures for the competency assessment of testing personnel.

In Australia, the Australian training authority has developed training packages for laboratory operations under which competency standards have been prepared [13]. In this competency's standards, there are seven domains broadly listed as collecting and analyzing data; communicating ideas and Information; planning and organizing activities; working with others and teams; using mathematical ideas and techniques, problem solving; and using technology.

Similarly, the Canadian Society for Medical Laboratory Science(CSMLS) has developed a competency profile for general medical laboratory technologist [14]. This profile has identified eight categories: safe work practices; data and specimen collection and handling; analytical processes; interpretation and reporting of results; quality management; critical thinking; communication and interaction; and professional practice.

In 2012, the Government of India provisioned laboratory services at primary health centers [15], in which 13 tests were included under the essential and desirable category). The Indian Council for Medical Research (ICMR) 2019 proposed to the Government of India to revise the test list for primary health centers and health and wellness centers at the helm of the comprehensive primary care rollout, which the Government of India accepted and provisioned in 2022 [16]. The revised list provisions sixty-four tests under seven categories, like hematology; clinical pathology; biochemistry; microbiology; specific diseases; other diagnostic tests; radiology.

MLTs are crucial in conducting tests at primary health centers, which are only available in 35.80% of primary health centers in India [17]. The competence of MLTs is essential for the overall performance of these centers, as laboratory services contribute significantly.

Due to the double burden of disease and increasing susceptibility to infections, most countries have adopted comprehensive primary care service packages, which include laboratory and diagnostic Services. Workforce competency is the driver of ensured efficiency and quality of the services. Competency assessment is the first step to improving the degree of workforce competency. There is no competency assessment framework for MLT's competency in the Indian primary healthcare setting. Hence, this study will help in bridging this gap.

## Study objective

The objective of this study was "to develop the competency assessment framework to assess the in-service Medical Laboratory Technologist's competency in primary healthcare Settings." The specific objective of this study was to identify the Medical Laboratory Technologists' competencies and to develop competency assessment tools for in-service MLTs in primary healthcare settings.

## Methods

### Study design

We used a five-stage consultative approach (Fig 1) to develop the MLTs competency assessment framework. Stage-by-stage specifics are as follows.

**Stage 1 desk review.** In this stage, we reviewed the global and Indian literature and identified 12 publications, including four from India, for detailed review. All these publications were applicable to primary healthcare settings across the globe.

**Stage 2 expert consultation.** In the second stage, we organized expert consultation to build consensus among MLT domains and competencies. This consultation was held on 7th april 2022.

**Stage 3 Competency Assessment Tool (CAT) development.** In this stage, we developed the instruments to assess the competencies in terms of knowledge, attitude, and skills.

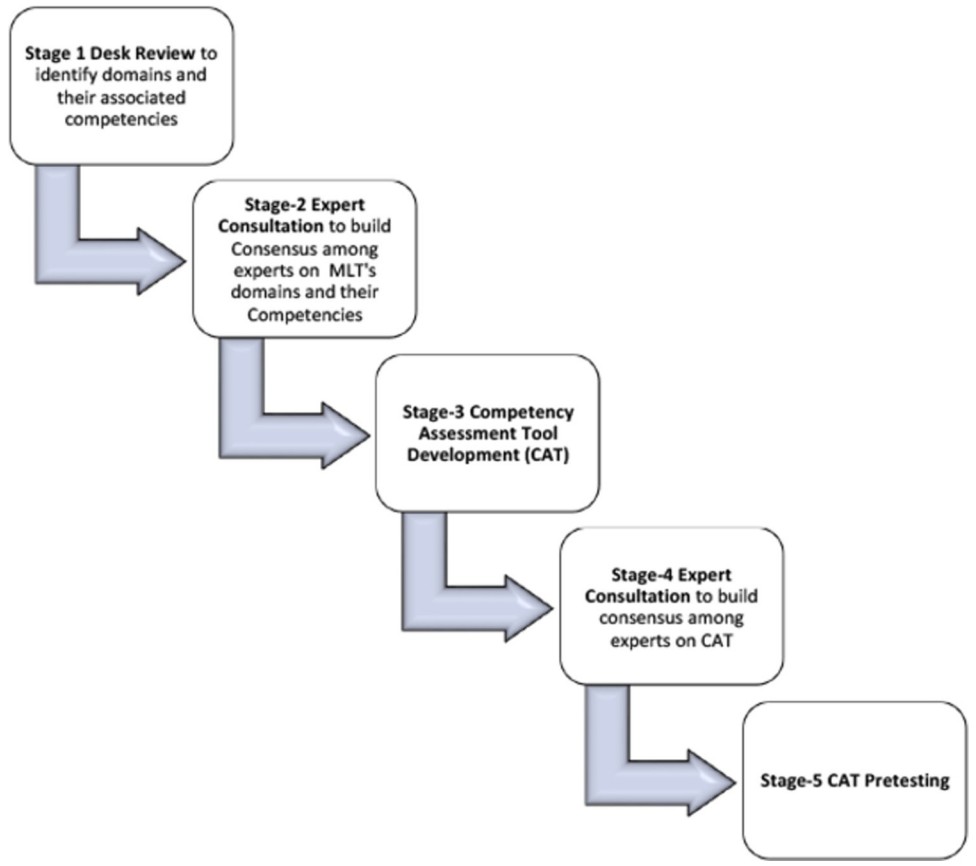

**Fig 1. Study stages.**

**Stage 4 expert consultation.** In this phase, we organized an expert consultation on 18[th] November 2022 to build consensus among experts on the developed CAT.

**Stage 5 CAT pre-testing.** We converted the final version of the Competency Assessment Tool into bilingual language, i.e., English and Odia (local language). Afterward, we pre-tested CAT inone of the public primary health care facilities in Odisha, on 13[th] January 2023.

**Experts.** During two rounds of expert consultations, 18 experts participated. All the experts were trained in medical laboratory science/pathology/laboratory medicine and have been involved in delivering medical laboratory services or as academia in medical laboratory science or involved in the process of policymaking as a multi-lateral organization representative or professional body representative or policy maker with over 20 Years of experience.

## Results

### Stage-1 competencies identification

Through the literature review, we identified 86 Medical Laboratory Technologist's competencies under 11 domains Table 1. These domains are safe work practices, data collection and specimen procurement/ receipt, specimen preparation and pre-analytical processing, equipment, instruments and reagents, assessment and analysis, recording and reporting, infection control, quality management, critical thinking, communication and interaction, and professional practice. Further, to refer to the detailed domain-wise competencies list, please refer to the S1 File.

### Stage-2 expert consultation

The virtual expert consultation was organized to build consensus among experts on MLTs' domains and their competencies. During the consultation, consensus was achieved on ten domains of MLTs in primary healthcare settings, Table 2. After that, competencies were discussed, and a consensus was made among the experts, S2 File.

**Domain-1. Human values and professional ethics.** In this domain, the experts arrived at a consensus on nine competencies Table 3. For complying with human values and professional ethics, one of the MLT's vital competencies is understanding the human values and ethics in a clinical laboratory. Apart from maintaining the confidentiality of health care information, MLTs must be able to demonstrate responsibilities towards patients, their attendants, regulatory bodies, and the profession. Respecting the diversity, dignity, and values of patients/clients and colleagues, contributing towards continual improvement in professional services, and

**Table 1. MLTs' competency domains.**

| Domains | References |
| --- | --- |
| Safe Work Practices | [2, 13, 18–27] |
| Data Collection and Specimen Procurement/ Receipt | [13, 23, 24, 26] |
| Specimen Preparation and Pre-analytical Processing | [13, 20, 22–24] |
| Equipment, Instruments, and Reagents | [2, 13, 18, 20, 22–27] |
| Assessment and Analysis | [13, 18, 22–24] |
| Recording and Reporting | [2, 13, 18, 20–25, 27] |
| Infection Control | [13, 22–25] |
| Quality Management | [2, 13, 18–27] |
| Critical Thinking | [24] |
| Communication and Interaction | [13, 21–27] |
| Professional Practice | [13, 22–24, 26, 27] |

**Table 2. MLTs' competency domains.**

| Medical Laboratory Technologist's Domains in Indian Primary Health Care Setting |
| --- |
| 1. Human Values and Professional Ethics |
| 2. Quality Management |
| 3. Communication |
| 4. Critical Thinking |
| 5. Equipment Instruments and Consumables |
| 6. Test Requisitions Data and Sample Collection |
| 7. Specimen Preparation |
| 8. Assessment & Analysis |
| 9. Recording and Reporting |
| 10. Laboratory Safety and Laboratory-Acquired Infection Control |

complying with legislation governing medical laboratories at the state and national level are essential competencies for MLTs. For professionals, it is also critical to recognize their own competency limitations and seek action to resolve those. In the laboratory setting, obtaining informed consent before the procedure wherever required and respecting patients' right to refuse is critical. Continuing education and training is one of the vital competencies for the MLTs' to learn and improve the quality of services offered at work environment.

**Domain-2 quality management.** Consensus was reached among experts on seven competencies, Table 4. Understanding and demonstrating the concept of quality management system including quality control, quality assurance, quality improvement, consistency, reproducibility, turn around time, and understanding the concept of corrective actions, root cause analysis, and preventive actions in case of any incidents/accidents and observable quality degradation during the pre-analytical, analytical, and post-analytical phase of testing are critical competencies for MLTs. For ensuring quality, it is critical to follow the established protocols as defined in the quality policy, process, and procedure manuals, and it is equally important that MLTs should be able to use simple mathematics to monitor and track the acceptability of quality control results and identifies documents, and reports deficiencies that may affect the testing quality and the generated test reports. Laboratory instruments are the backbone of any laboratory; hence, MLTs must understand and perform preventive maintenance and calibration of those instruments referring to established protocols. To ensure quality in the laboratory, it is also important for MLTs to demonstrate knowledge about inventory control by adopting the First

**Table 3. Human values and professional ethics- competencies.**

| |
| --- |
| 1. Able to understand the human values and ethics in a clinical laboratory. |
| 2. Able to maintain the confidentiality of healthcare information. |
| 3. Able to demonstrate their responsibilities towards patients, their attendants (public), regulatory bodies, the profession, and himself/herself as expected from medical laboratory professionals |
| 4. Respect the diversity, dignity, values, and beliefs of patients/clients and colleagues. |
| 5. Able to contribute towards continual improvement in the professional services, focussing on essential aspects like quality of work. |
| 6. Able to comply with legislation that governs medical laboratory technology at local, state, or national levels as applicable. |
| 7. Able to recognize limitations of own competency and seek action to resolve. |
| 8. Able to obtain informed consent before the procedure wherever required and respect a patient's right to refuse. |
| 9. Able to develop his/ her interest to participate in continuing education and training–Interest to learn and improve the quality of the work environment. |

**Table 4. Quality management.**

| |
|---|
| 1. Able to understand and demonstrate the concepts of the quality management system, including quality control, quality assurance, quality improvement, consistency, reproducibility, turn around time, etc., |
| 2. Able to understand corrective actions, root cause analysis, and preventive actions in case of any incidents/accidents and observable quality degradation during the pre-analytical, analytical, and post-analytical phases of testing. |
| 3. Able to follow established protocols as defined in the quality policy, process, and procedure manuals. |
| 4. Able to use simple mathematics to monitor and track the acceptability of quality control results. Identifies, documents, and reports deficiencies that may affect the testing quality and the generated test report. |
| 5. Able to understand and perform preventive maintenance and calibration of Laboratory instruments according to established protocols. |
| 6. Able to participate in internal and external quality assurance activities, e.g., audits accreditation. |
| 7. Able to demonstrate knowledge of inventory control by adopting FIFO (First in and First Out) |

In and First Out (FIFO) method and participating in quality assurance activities, internal and external, like audits and accreditation.

**Domain-3 communication.** Experts in this domain agreed on five competencies, Table 5. For delivering effective quality laboratory services, MLTS need to communicate effectively with patients/clients, colleagues, and other health care professionals in local or regional language with respect to: active listening; verbal communication; non-verbal communication; written communication; conflict management; identifying barriers to effective communication; using technology appropriately to facilitate communication. In medical laboratory functioning, medical terminologies are being used; hence MLTS need to understand the medical terminologies and the abbreviations used. Delivering health services is teamwork, where MLTs need to demonstrate effective interdisciplinary /intra-professional team skills through communication, collaboration, and role clarification. Demonstration of adaptive skills during interacting with patients is also an important competency for MLTs.

**Domain 4. Critical thinking.** In this domain, consensus among experts was reached on four competencies, Table 6. For MLTs, it is important to engage in reflective practice; consciously analyses, make decisions, and draw conclusions to improve future practice. It is important for MLTs to accommodate valid priorities and ensure efficient use of 4 M (Money,

**Table 5. Communication.**

| |
|---|
| 1. Able to communicate effectively with patients/clients, colleagues, and other health care professionals in local or regional language with respect to: |
| a. Active listening |
| b. Verbal communication |
| c. Non-verbal communication |
| d. Written communication |
| e. Conflict management |
| f. Identifying barriers to effective communication |
| g. Using technology appropriately to facilitate communication. |
| 2. Able to understand medical terminology used at least related to the functioning of medical laboratories |
| 3. Able to understand and use the abbreviations commonly used in medical terminology. |
| 4. Able to demonstrate effective interdisciplinary/intra-professional team skills through: |
| (i) Communication |
| (ii) Collaboration |
| (iii) Role clarification |
| 5. Able to demonstrate adaptive skills when interacting with patients |

**Table 6. Critical thinking.**

| |
|---|
| 1. Able to engage in reflective practice; consciously analyses, makes decisions and draws conclusions to improve future practice. |
| 2. Able to organize work to accommodate valid priorities. |
| 3. Able to ensure efficient use of 4 M (Money, Man, Material, and Minutes). |
| 4. Able to demonstrate effective problem-solving/trouble-shooting strategies and initiate the appropriate follow-up. |

Man, Material, and Minutes). For delivering efficient laboratory services, it is also critical for MLTs to demonstrate effective problem-solving/trouble-shooting strategies and initiate the appropriate follow-up.

**Domain 5. Equipment/instruments and consumables.**   Experts were in consensus for 13 competencies in this domain, Table 7. Under the equipment/instruments and consumables domain, one of the competencies is to prepare reagents, calibrators/standards, and quality control materials and to demonstrate the shelf life of ready-to-use commercially procured and in-house prepared materials and reagents and check the functional quality of all reagents / diagnostic kits/cards/ chemicals used. For any laboratory, it is critical to use positive and negative control for all tests; hence, one of the critical competencies for MLTs is to demonstrate positive and negative control in all tests. For optimum use of laboratory instruments/equipment, it is important for MLTs to demonstrate the work principles of instruments. It is equally important for the laboratory professional to perform risk assessment while using any laboratory instruments. Knowledge about the calibration concepts of equipment and its importance in laboratory services and maintenance for equipment working and its integrity through regular maintenance are vital competencies. Labeling the equipment/instrument with their respective unique I.D.s, date of purchase, date of installation, date of putting into service, date of the last calibration, and name & contact of mechanic whom to inform in case of emergency and its maintenance of recording of all equipment from time to time is also important for MLTs. It is also important for the laboratory professional to demonstrate functional quality checks of all the available equipment/instruments. It is also important for the laboratory professional to

**Table 7. Equipment instruments and consumables.**

| |
|---|
| 1. Able to prepare reagents, calibrators, standards, and quality control materials. |
| 2. Able to demonstrate the shelf life of ready-to-use commercially procured and in-house prepared materials and reagents. |
| 3. Able to check the functional quality of all reagents / diagnostic kits/cards/ chemicals used in the laboratory. |
| 4. Able to use/demonstrate positive and negative controls in all tests. |
| 5. Able to demonstrate the work principles of all the laboratory instruments/equipment. |
| 6. Able to perform risk assessment while using any equipment. |
| 7. Knowledgeable about the calibration concepts of equipment and its importance in laboratory services. |
| 8. Able to maintain equipment working and its integrity by its regular maintenance. |
| 9. Able to inform competent authorities if the equipment has gone out of order. |
| 10. Able to maintain maintenance records of all equipment from time to time. |
| 11. Able to demonstrate functional quality checks of all the available equipment/instruments. |
| 12. Able to label the equipment/instrument with their respective unique I.D.s, date of purchase, date of installation, date of putting into service, date of the last calibration, and name and contact of address mechanic whom to inform in case of emergency. |
| 13. Able to prepare and demonstrate SOPs for the use of all equipment/instruments, test procedures, and their display on workbenches |

prepare and demonstrate SOPs for using all equipment/instruments, test procedures, and their display on workbenches.

**Domain 6. Test requisition data and sample collection.** In this domain, consensus was reached on ten competencies among experts, Table 8. Understanding the relevant information provided for test requests on requisition forms, mainly patient identification, and performing venepuncture and capillary blood collection to obtain appropriate samples for laboratory analysis are critical competencies for MLTs. Apart from patient counselling/ preparation for the collection of clinical samples, lab professionals must provide all relevant information about precautions on specimen collection, transportation, and storage to the patients/clients wherever required. Demonstrating types and using of procedures or tools for sample collection and sample type that needs to be collected for a required test, like serum plasma or whole blood, etc., are vital competencies for MLTs. Professionals should also know about the anticoagulants used for blood or other samples if required. For laboratory professionals, it is also important to demonstrate skin disinfection procedures that are used before taking any blood sample and should be able to perform the sample collection for tests done at primary health centres and chain of custody procedures relating to specimens as per requirements of the test procedure. Professionals should also be able to pack the samples safely and transport clinical samples to higher competent laboratories for advanced testing or any other purpose, considering all sample and environmental safety requirements.

**Domain 7. Specimen preparation.** Experts arrived at a consensus on three competencies in this domain, Table 9. Processing specimens considering identified priority and verifying that the pertinent specimen data and test requisition form are critical competencies for laboratory professionals. It is also important for the professional to assess the specimen's suitability for testing and prepare specimens for analysis like blood, body fluids, and other clinical specimens for any examination, including microscopic.

**Domain 8 assessment and analysis.** In this domain, consensus arrived for 14 competencies among the experts, Table 10. For MLT, following the test/equipment's SOPs properly and understanding their importance is important. For staining, it is critical for laboratory professional to apply the physical and chemical principles of staining and the quality of staining and initiates corrective action. Microscopy is one of the procedures widely used in primary health care settings; hence, lab professionals need to be able to apply routine microscopy and to be

**Table 8. Test requisition data and sample collection.**

| |
|---|
| 1. Able to understand relevant information provided for test requests on requisition forms, which is essential for the interpretation of results or critical alerts or recording purposes. |
| 2. Able to confirm the identity of the patient and perform venipuncture and capillary blood collection to obtain appropriate samples for laboratory analysis. |
| 3. Able to provide all relevant information about specimen collection, transportation, and storage precautions to the patients/clients wherever required. |
| 4. Able to perform patient counseling/preparation for collection of clinical samples. |
| 5. Able to demonstrate the types and use of procedures or tools for sample collection, like vacutainers for blood samples. |
| 6. Able to demonstrate what type of sample is to be collected for a required test like serum plasma or whole blood, etc. |
| 7. Should know what types of anticoagulants are to be used for blood or other samples if required. |
| 8. Able to demonstrate skin disinfection procedures before taking any blood sample or otherwise wherever required. |
| 9. Able to perform the sample collection for tests routinely done at PHCs and chain of custody procedures relating to specimens as per requirements of the test procedure. |
| 10. Able to pack the samples safely and transport clinical samples to higher competent laboratories for advanced testing or any other purpose, keeping in view all sample and environmental safety requirements |

**Table 9. Specimen preparation.**

| |
|---|
| 1. Able to identify and process specimens considering priority and able to verify the pertinent data on the specimen and test requisition form. |
| 2. Able to assess the specimen's suitability for testing. |
| 3. Able to prepare specimens for analysis like blood, body fluids, and other clinical specimens for any examination, including microscopic. |

able to apply the principles of light measuring systems used in common instruments: reflectometry turbidometry. For test result assessment and analysis, it is important for the professional to be capable of assessing results, identifying sources of interference, and initiating corrective action. Considering the usage of point-of-care tests, it is also important for lab professionals to perform card-based immunoassay tests using commercially available ready-to-use kits for tests done at PHCs with Knowledge of all the precautions affecting the test results. For using analyzers/ semi-auto analyzers, MLTs should be able to apply the principles of commonly used analyzers/ semi-auto analyzers for hematology and biochemistry. Manual counting procedures using cell counters professionals should certainly be able to perform. It is also important for MLTs to apply the principles of hemostasis to perform coagulation testing. For tests under biochemistry, professionals should be able to perform through manual methods, too. While in microscopy, he/she should be able to identify and evaluate the morphology of cellular and non-cellular elements in microscopic preparations. Understanding the differentiation between clinically significant and insignificant findings in the test done at the laboratory is one of the critical competencies for professionals. For MLTs, performing the point of care testing and assessing results is also important. While performing laboratory processes for any tests, the professional should adhere to guidelines for specimen retention, storage, transportation, and disposal.

**Domain 9. Recording and reporting.** In this domain, experts arrived at the consensus for nine competencies, Table 11. For MLTs, It is a critical competency to report results of all types of tests done in PHCs based on manual reporting like; microscopy, colourimetry, strip tests/ card tests, etc., meeting quality control criteria. For reporting results, it is also essential to

**Table 10. Assessment and analysis.**

| |
|---|
| 1. Able to follow the test/equipment SOPs properly and understand their importance. |
| 2. Able to apply the physical and chemical principles of staining and the quality of staining and initiate corrective action. |
| 3. Able to apply the principles of routine microscopy. |
| 4. Able to apply the principles of light measuring systems used in common instruments: reflectometry turbidometry. |
| 5. Able to assess results, identify sources of interference, and initiate corrective action. |
| 6. Able to perform card-based immunoassay tests and tests using commercially available ready-to-use kits for tests done at PHCs with Knowledge of all the precautions affecting the test results. |
| 7. Able to apply principles of commonly used analyzers/ semi-auto analyzers for hematology and biochemistry if available at respective PHC. |
| 8. Able to perform manual counting procedures using cell counters, etc. |
| 9. Able to apply the principles of hemostasis to perform coagulation testing. |
| 10. Able to perform some common biochemistry tests using manual methods. |
| 11. Able to Identify and evaluate the morphology of cellular and non-cellular elements in microscopic preparations. |
| 12. Able to differentiate between clinically significant and insignificant findings to the test done at the laboratory. |
| 13. Able to perform point-of-care testing and assess results. |
| 14. Able to adhere to guidelines for specimen retention, storage, transportation, and disposal |

**Table 11. Recording and reporting.**

| |
|---|
| 1. Able to report results of all types of tests done in PHCs based on manual reporting like microscopy, colorimetry, strip tests/ card tests, etc., meeting quality control criteria. |
| 2. Able to understand the standard units used in test reports as agreed between laboratory professionals and clinicians. |
| 3. Should know the importance of referring to any preliminary report of the same test on the same patient. |
| 4. Able to compare the results with positive and negative controls. |
| 5. Able to record the test results in the Laboratory Information System or on registers as applicable. |
| 6. Able to keep records/reporting data in safe custody. |
| 7. Able to recognize and act on critical values by timely communicating the critical reports to the concerned physician. |
| 8. Able to release a report within established TAT (Turn Around Time) |
| 9. Able to confirm that the laboratory's report distribution/delivery system is efficient and that the report reaches the clinician or patient confidentially and on time |

understand the standard units used in test reports as agreed between laboratory professionals and clinicians. Results comparison with positive and negative control improves the report quality. When referring to any preliminary report on the same test on the same patient, the professional should understand its importance. Recording test results is critical; hence, the professional must know how to feed results data into the laboratory information system or on registers, as applicable. In the case of registers, it is one of the MLT's responsibilities to keep records in safe custody. It is also vitally important for the professional to recognize and act on critical values by timely communicating the critical reports to the concerned physician. Releasing the test report within the established turn around time is essential for the professional, and they also need to confirm that the laboratory's report distribution/delivery system is efficient and that the report reaches the clinician or patient confidentially on time.

**Table 12. Laboratory safety and laboratory-acquired infection control.**

| |
|---|
| 1. Able to demonstrate general and specific safety precautions of clinical laboratory |
| 2. Able to use personal protective equipment appropriately, e.g., gloves, gowns, masks, face shields, and aprons. |
| 3. Able to know about laboratory hygiene and infection control practices / Policies. |
| 4. Able to minimize possible dangers from biological specimens and use biosafety equipment. |
| 5. Able to use laboratory safety devices, e.g., safety pipetting devices, safety containers, and carriers. |
| 6. Able to label, date, handle, store, and dispose of chemicals, dyes, reagents, and solutions according to legislation, e.g., Master Safety Data Sheet (MSDS) |
| 7. Able to handle and disposes of sharps (Bio-medical waste policy) with special references to syringes and their needles. |
| 8. Able to store, handle, transport, and dispose of biological and other hazardous materials according to the Bio-medical waste policy |
| 9. Able to use disinfection and sterilization methods to disinfect materials to be used or disposed of. |
| 10. Able to select disinfectant as required. |
| 11. Able to test the efficacy of Sterilizers. |
| 12. Able to document all incidents related to safety and personal injury like Needle Stick Injury |
| 13. Able to apply the standard precautions to prevent the spread of infection as per organization requirements / local rules and other rules as applicable. |
| 14. Able to minimize contamination of materials, equipment, instruments, and environment by aerosol and splatter. |
| 15. Able to follow protocols for care following exposure to blood or other body fluids as required. |
| 16. Able to place appropriate signs wherever and whenever required, e.g., hazardous, flammable, restricted entry, containment zone etc. |
| 17. Able to maintain hand hygiene by washing hands before and after patient contact and after any activity likely to cause contamination. |
| 18. Able to use alcohol-based hand sanitizers, if justified, e.g., soap and water are unavailable or any other such barrier. |
| 19. Able to deal with sharp cuts and abrasions |

**Domain 10. Laboratory safety and laboratory-acquired infection control.** Experts agreed on 19 competencies in this domain, Table 12. For MLTs, it is one of the critical competencies to demonstrate the general and specific safety precautions of clinical laboratories. Apart from demonstrating the general and specific, it is also important for them to know about laboratory hygiene and infection control practices /policies and to use personal protective equipment appropriately, e.g., gloves, gowns, masks, face shields, and aprons. Professionals should also be able to use laboratory safety devices, e.g., safety pipetting devices, safety containers, and carriers, and be able to minimize possible dangers from biological specimens and use biosafety equipment. Labeling date, handle, store, and dispose of chemicals, dyes, reagents, and solutions according to legislation, e.g., Master Safety Data Sheet (MSDS) and handling and disposing of sharps, particularly syringes and needles, MLTs should essentially be competent. In complying with biomedical waste management policy, professionals should be able to store, handle, transport, and dispose of biological and other hazardous materials, select appropriate disinfectants, and use disinfection and sterilization methods to disinfect materials. The efficacy of sterilizer play a significant role in complying with quality compliance; hence, efficacy testing for sterilizer, professional should undoubtedly be handy. In the laboratory, it is not uncommon to experience needle stick injury; hence, documenting all the incidents related to safety and personal injury like needle stick injury is essentially warranted by laboratory professionals. Even in the case of infection or contamination, the laboratory professional must minimize contamination of materials, equipment, instruments, and environment by aerosol and splatter. In between, it is essentially warranted in the laboratory to follow protocols for care following exposure to blood or other body fluids as required. It is important for MLTs to apply standard precautions to prevent the spread of infection as per organization requirements / local rules and different rules as applicable. Appropriate signage is critical for any laboratory; hence, the professional must ensure the appropriate signage wherever and whenever required, e.g., hazardous, flammable, restricted entry, containment zone etc., in the laboratory. In the event of sharp cuts and abrasions, professionals should be competent to deal with them. Hand hygiene is one of the critical practices that need to be followed in the laboratory; hence, professionals should maintain hand hygiene by washing hands before and after patient contact and after any activity likely to cause contamination. In the case of a soap supply shortage, S/he should use alcohol-based hand sanitizers.

Based on the experts inputs during the consultation 11 domains were restructured into 10. For example Safe work practices and infection control is considered as single domain. Under assessment and analysis, several competencies were removed as they are not applicable in the primary healthcare settings in India (Refer S2 File).

## Stage-3 Competency Assessment Tool (CAT) development

Based on the literature, we developed a Competency Assessment Tool, S3 File, to assess Medical Laboratory Technologist's competency in terms of knowledge, attitude, and skills. To assess knowledge and attitude, we developed a questionnaire in which 138 questions were constructed, Table 13. Of these, 89 questions were for knowledge, and the remaining 49 were for attitude. Further, we used three instruments to assess MLTs skills: observational, mini clinical laboratory evaluation, and case studies [28, 29]. Under observation, we identified 56 skills, 13 skills under mini-clinical exercises, and five as case study exercises. We developed the tracer elements for observation and mini-clinical evaluation exercises for all the skills and listed those tracer elements under means of verification. Further, we developed the standard score based on means for verification for the skills under observation and mini-clinical evaluation

**Table 13. Competency assessment tool-instrument details.**

| Competency Elements | Instruments | Number of Questions/ Skills | |
|---|---|---|---|
| | | **Pre-Consultation** | **Post Consultation** |
| **Knowledge** | Questionnaire | 89 | 53 |
| **Attitude** | Questionnaire | 49 | 23 |
| **Skills** | Observational Checklist | 56 | 26 |
| | Mini Clinical Evaluation Exercises | 13 | 4 |
| | Case Study Exercises | 5 | 4 |

exercises; we developed the standard score definition, Table 14. Please refer S3 File for this version of CAT.

## Stage 4 expert consultation

In this stage, to develop a consensus among experts on the MLT's competency assessment tool, we organized an expert consultation on 18[th] November 2022 in Bhubaneswar, Odisha, India. The consensus among experts arrived at 76 questions, of which 53 were for knowledge and the rest 23 were for attitude. Under skill assessment, consensus was achieved among experts for 26 skills under Observational exercises, four for mini-clinical, and four for case studies, Table 14. Please refer the S4 File for this version of CAT.

## Stage-5 CAT pre-testing

In this stage, we converted the English version of CAT into a bilingual format in English and Odia (vernacular language). After that, we administered the bilingual CAT with one of the Medical Laboratory Technologist in one of the public primary health care facilities in Khorda district, Odisha, India. Based on the pre-testing learnings, we improvised the CAT and finalized it as a bilingual Competency Assessment Tool, supporting information file. For section-wise learning details, please refer to Table 15.

Three assessors, all graduates in medical laboratory science, took part in the pre-testing. One assessor focused on establishing a rapport with the facility administrator and assessing skills through observation. Meanwhile, the other two assessors evaluated knowledge, attitude, and skills through mini clinical evaluation exercises and case study exercises. The average time spent on administering the questionnaire, observational checklist, mini clinical exercises, and case studies was 1.5 minutes, 3 minutes, 10 minutes, and 3 minutes, respectively (Table 16). Mainly, assessments were conducted verbally, except for observation and mini clinical exercises.

**Table 14. Score definition.**

**Score Definition**
 **A. None**—No demonstrated skills at all/does not perform the task(s) completely.
 **B. Limited** Demonstrated very limited strengths/skills in this area.
 **C. Some**—Demonstrated some ability/skills in this area.
 **D. Strong**—Demonstrated strong skills/strength in this area.
 **E. Excellent**—Demonstrated excellent skills/strength in this area.
 **F. Not applicable**
 **G. Don't know**- Not even heard about that skill.
 **H. Skill limitation** is related to resource limitations.

**Table 15. Pre-testing learning.**

| CAT Section | Sr. No. | Learnings |
|---|---|---|
| C1. Questionnaire (Knowledge and Attitude) | 4 | Respondents were not clear with the word **critical components**. So, to improve the clarity, we gave some examples for the critical component, like **addressing the complaints and feedback from Patients, Physicians, and Staff and learnings from accidents/ Incidents.** |
| | 45 | Respondents were not very clear with the term-transcript check. To improve clarity, we extended the question with one example for transcript check as—One should check and observe whether the rough data of results, the data entered into the lab register or on LIS, and the report data entered into the final report format to be given to the patient are same. |
| C2. Observational Tools | 6 | We realized the unavailability of the listed equipments for efficacy testing. Hence, to accommodate this resource limitation, we extended the list of equipment to **any other instruments/equipment used in the laboratory**. |
| | 7 | First Entry and First Out (FEFO) is equally important as First in and First Out. So, we extended FI-FO with **FEFO.** |

## Discussion

We developed the competency assessment framework for In-Service Medical Laboratory Technologists in primary healthcare settings. Under this, we identified the MLTs' role-based domains and their competencies. Along with the domains and their competencies, we developed the Competency Assessment Tool (CAT) to assess the level of competencies among in-service MLTs. As CAT will help identify the competency gap, this framework will help the policymakers and implementers design solutions to bridge the competency gap among the in-service MLTs in primary healthcare settings. It will further lead to improved MLT performance in primary healthcare.

### Factors influencing MLTs competency

Pre-Service Education influences the MLTs' competency the most. It is facing multi-pronged challenges even though entry-level qualifications are different at different levels; there is no uniformity in training periods and no standard curriculum [30]. Apart from these, another pressing challenge for teaching institutions is the shortage of qualified faculty. In India, MLTs and other allied health professionals are still not regulated. However, In 2021, the Government of India came up with the National Commission for Allied and Healthcare Professions Act to regulate and maintain allied health professional's education standards and services, which is yet to be implemented in most states [31, 32].

In the Indian public health system, the union and state governments have made the limited efforts to capacitate the Medical Laboratory Technologists. Government agencies efforts are

**Table 16. CAT section wise average time and asssessment mode.**

| Competency Elements | Instruments | Numbers | Average time | Assessment Mode |
|---|---|---|---|---|
| **Knowledge** | Questionnaire | 53 | 1.5 Min. | Verbal |
| **Attitude** | Questionnaire | 23 | 1.5 Mins. | Verbal |
| **Skills** | Observational Checklist | 26 | 3 Mins | Observation |
| | Mini Clinical Evaluation Exercises | 4 | 10 Mins | Observation and Verbal |
| | Case Study Exercises | 4 | 3 Mins | Verbal |

primarily fragmented in building LTs capacity under the Revised National Tuberculosis Control Program, National Leprosy Eradication Programme, and others [33, 34].

### Importance of competency assessment framework in developing competency-based training for medical laboratory technologists

It is evident that the Competency framework for training helps improve MLTs performance irrespective of the type of settings [35]. In India, there are lacunae in providing training to the in-service workforce. Competency-based Training (CBT), a structured training and assessment system that allows individuals to acquire skills and knowledge to perform work activities to a specified standard, is a globally accepted solution to capacitate the in-service professional [36]. The competencies and the competency assessment tool will play a foundational role in developing the Competency-based training manual and other resource materials for MLTs in primary healthcare Settings.

## Strengths and limitations

The strength of this study is the consultative approach through which this MLT competency assessment framework has been developed. Eighteen experts in medical laboratory science participated in this process. These experts have over 20 Years of experience in teaching, policy-making, and practice from teaching institutions, union, state governments, professional bodies, and private and public laboratory service providers. In both consultations, 18 experts participated; hence, we recognize that the experts who participated in this study cannot be considered the complete representation of all the stakeholders for the in-service MLTs competency development initiative.

Apart from that, we did not use any tool to check the validity of the score definition for Skills assessment (observation and mini clinical exercises). We also understand that we pre-tested CAT in only one public facility in Odisha, India, and did not use any quantitative tool to check its validity. Another limitation is the variety of regional languages, which necessitates pre-testing the CAT in the regional language in order to adapt it to the specific requirements of the region/state.

The developed competency assessment framework is comprehensive. Hence, administering CAT to assess in-service MLT competency is time-consuming. As the role of medical laboratory professionals evolves based on advancement in medical laboratory science, this competency assessment framework would also need periodic review and updating.

## Conclusion

The framework for MLTs competency assessment aims to pinpoint the training requirements for MLTs working in primary health care settings. Its implementation can assist in creating competency-based training programs in primary healthcare facilities, leading to the upskilling of MLTs and the enhancement of laboratory service quality in Indian primary healthcare.

### Practice implication

The MLTs competency assessment framework enables health system implementers to identify gaps in the competencies of MLTs working in primary health care settings. Subsequently, they can create and design competency-based training programs to enhance their skills and abilities. It can also contribute to developing the curriculum and pedagogy of education of MLTs.

The indian public health systems have a structured framework in place to strengthen the skills of in-service personnel, including MLTs. The allocated resources for this capacity-

building initiative can be utilized to cover the expenses associated with assessments and the development of competencies.

## Supporting information

**S1 File. Literature based medical laboratory technologist's competencies.**
(PDF)

**S2 File. Competencies for medical laboratory technicians in primary health care setting.**
(PDF)

**S3 File. Competency assessment tool-medical laboratory technologist-lit based.**
(PDF)

**S4 File. Competency assessment tool-medical laboratory technician-post consultation.**
(PDF)

**S5 File. Competency assessment tool-medical laboratory technician-bilingual.**
(PDF)

## Acknowledgments

We would like to thank Mr. Rajeev Sadanandan, Dr. Kumaravel Ilangovan, from HSTP and Mrs. Shivangini Kar Dave from Novo Nordisk Foundation for their useful suggestions on development of this paper. We also recognize the contributions of the experts who participated in the consultations during the study.

## Author Contributions

**Conceptualization:** Sanjeev Kumar.

**Data curation:** Sanjeev Kumar.

**Methodology:** Sanjeev Kumar.

**Project administration:** Sanjeev Kumar.

**Validation:** Sanjeev Kumar.

**Writing – original draft:** Sanjeev Kumar.

**Writing – review & editing:** Sanjeev Kumar, Gaurav Chhabra, Kaptan Singh Sehrawat, Malkit Singh.

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
