## [Decision Letter · Decision Letter 0]

28 Dec 2023

PONE-D-23-32693Developing a Competency Assessment Framework for Medical Laboratory Technologists in Primary Health Care Settings in IndiaPLOS ONE

Dear Dr. Kumar,

Thank you for submitting your manuscript to PLOS ONE. After careful consideration, we feel that it has merit but does not fully meet PLOS ONE’s publication criteria as it currently stands. Therefore, we invite you to submit a revised version of the manuscript that addresses the points raised during the review process.

Please respond to the helpful comments the reviewers have made. If there has not been as yet any implementation of this framework, please comment based on previous experience elsewhere, or experience with implementing other frameworks in India.

We look forward to receiving your revised manuscript.

Kind regards,

Susan Horton

Academic Editor

PLOS ONE

2. Please include a caption for figure 1.

Additional Editor Comments:

Please consider the comments made by the reviewers. It may not be easy to discuss the experience of implementation of the framework if it has not yet been implemented, but please comment based on any experience that is available.

Reviewers' comments:

Reviewer's Responses to Questions

**Comments to the Author**

1. Is the manuscript technically sound, and do the data support the conclusions?

Reviewer #1: Partly

Reviewer #2: Yes

2. Has the statistical analysis been performed appropriately and rigorously? 

Reviewer #1: N/A

Reviewer #2: N/A

3. Have the authors made all data underlying the findings in their manuscript fully available?

Reviewer #1: No

Reviewer #2: Yes

4. Is the manuscript presented in an intelligible fashion and written in standard English?

Reviewer #1: Yes

Reviewer #2: Yes

5. Review Comments to the Author

Reviewer #1: This is a detailed report of the steps taken to create a comprehensive competency assessment framework for medical laboratory techologists working in primary healthcare settings in India. It would be good if the authors analyse and report its robustness when applied on the ground.

Reviewer #2: The manuscript presents a meaningful project to develop a Competency Assessment Tool (CAT) for MLT in public primary health care settings in India. Please address the following issues:

• Results and Discussion should include examples on objections or matters of concern raised against the competencies that have been proposed but did not reach unanimous consensus.

• Is there record regarding the (average) time spent on completing various parts of the CAT listed in Additional file 3, such as time spent on each questionnaire and each observational assessment? How many assessors were assigned on Observational Tools (Skills) and Mini Clinical Laboratory Skills? How are the case studies conducted, verbally or written?

• While such professional assessment is usually conducted by nominees working on voluntary basis, the exercise will still incur financial commitments. Please suggest plans for such financial support.

• What is the plan of implementing the CAT studied and will it become mandatory for MLT license or practise? Please also recommend if such assessment will be conducted regularly and the frequency if applicable.

• In addition to the limitations discussed near the end of the manuscript, are there other hurdles that the authors anticipate in the implementation of CAT established in this exercise? For instance, will the diversity of local languages be a problem? The CAT needs to be improvised when the bilingual version is finalized as described under Stage 5 Pre-testing.

6. PLOS authors have the option to publish the peer review history of their article (what does this mean?). If published, this will include your full peer review and any attached files.

Reviewer #1: No

Reviewer #2: No

---

## [Author Response · Author response to Decision Letter 0]

20 Jan 2024

01 Please consider the comments made by the reviewers. It may not be easy to discuss the experience of implementation of the framework if it has not yet been implemented, but please comment based on any experience that is available. 

Res- Thank you for your suggestions/comments. We have made the necessary changes.

02 Results and Discussion should include examples on objections or matters of concern raised against the competencies that have been proposed but did not reach unanimous consensus. Res-Thank you for your comments. We have made the necessary changes in the results. Please refer Page 20 and Line number 408 to 411.

03 Is there record regarding the (average) time spent on completing various parts of the CAT listed in Additional file 3, such as time spent on each questionnaire and each observational assessment? How many assessors were assigned on Observational Tools (Skills) and Mini Clinical Laboratory Skills? How are the case studies conducted, verbally or written? 

Res-Thank you. We have addressed your comments. Please refer Page 23 and Line number 448 to 457.

04 While such professional assessment is usually conducted by nominees working on voluntary basis, the exercise will still incur financial commitments. Please suggest plans for such financial support. 

Res- Thank you. We have addressed your comments. Please refer Page 26, Line No. 523 to 526

05 What is the plan of implementing the CAT studied and will it become mandatory for MLT license or practise? Please also recommend if such assessment will be conducted regularly and the frequency if applicable. 

Res- Thank you for the valid question. Policymakers have yet to make this policy decision to implement this framework. 

06 In addition to the limitations discussed near the end of the manuscript, are there other hurdles that the authors anticipate in implementing CAT established in this exercise? For instance, will the diversity of local languages be a problem? The CAT needs to be improvised when the bilingual version is finalized as described under Stage 5 Pre-testing. 

Res- Thank you. We have addressed your comments. Please refer Page 25 and Line Number 505 to 507

---

## [Editor Report · Decision Letter 1]

24 Jan 2024

PONE-D-23-32693R1Developing a Competency Assessment Framework for Medical Laboratory Technologists in Primary Health Care Settings in IndiaPLOS ONE

Dear Dr. Kumar,

Thank you for submitting your manuscript to PLOS ONE. After careful consideration, we feel that it has merit but does not fully meet PLOS ONE’s publication criteria as it currently stands. Therefore, we invite you to submit a revised version of the manuscript that addresses the points raised during the review process. The manuscript needs to be thoroughly proofread such that it is professional in style. See specific suggestions below.

We look forward to receiving your revised manuscript.

Kind regards,

Susan Horton

Academic Editor

PLOS ONE

Journal Requirements:

Additional Editor Comments:

I am satisfied with the response to the reviewers' comments. I am requesting that you carefully proofread the manuscript such that it is in professional form. Here are suggestions:

Line 72 (and elsewhere) – it is unusual to capitalize Clinically. Similarly, throughout the manuscript I think it is inconsistent to capitalize Competencies in some places but not others, unless referring to Competency Assessment Tool. Similarly Primary Care is not usually capitalized (throughout).

Lines 118-121: it would be better to either remove the capitalization on the various terms (or else to add capitals to planning (line 119) - consistency is important

Line 171: need space after “CAT” and before “in” and remove comma after “in”

Line 174: Medical Laboratory Services and Pathology are capitalized, but not laboratory science – be consistent! This inconsistency exists in several other places in the text where a list is provided, with some (but not all) of the items capitalized. In many cases, they would be better not capitalized unless referring to a proper noun.

Line 183: spelling of “domains”, and leave a space after “domains” and before “Table”

Line 390: spelling of “domains”

Line 392: suggest “they are” in place of “its is”

Line 424: remove space before “ .” and also add a space after “,”

P23, table 15: “list of equipment” not “equipments”

Line 476: delete “Care” after “healthcare”

Line 479: add space before “Eighteen”

---

## [Author Response · Author response to Decision Letter 1]

25 Jan 2024

As suggested, made the necessary changes.

---

## [Editor Report · Decision Letter 2]

30 Jan 2024

Developing a Competency Assessment Framework for Medical Laboratory Technologists in Primary Health Care Settings in India

PONE-D-23-32693R2

Dear Dr. Kumar,

We’re pleased to inform you that your manuscript has been judged scientifically suitable for publication and will be formally accepted for publication once it meets all outstanding technical requirements.

Kind regards,

Susan Horton

Academic Editor

PLOS ONE

Additional Editor Comments (optional):

Thanks for proofreading the manuscript. I have four small suggestions.

line 123: please retain the capitalization on Canadian Society for Medical Science - this is the proper name of the society

Line 131 - I think it is more appropriate to retain capitalization on Government of India

line 143: no need to capitalize Care

line 151 - can abbreviate MLT
---

## [Editor Report · Acceptance letter]

20 Mar 2024

PONE-D-23-32693R2 

PLOS ONE

Dear Dr. Kumar, 

I'm pleased to inform you that your manuscript has been deemed suitable for publication in PLOS ONE. Congratulations! Your manuscript is now being handed over to our production team.

Kind regards, 

on behalf of

Dr. Susan Horton 

Academic Editor

PLOS ONE